# Giant Panda Microhabitat Study in the Daxiangling Niba Mountain Corridor

**DOI:** 10.3390/biology12020165

**Published:** 2023-01-20

**Authors:** Wei Jia, Shasha Yan, Qingqing He, Ping Li, Mingxia Fu, Jiang Zhou

**Affiliations:** 1School of Life Sciences, Guizhou Normal University, Guiyang 550001, China; 2Key Laboratory of Animal Ecology and Conservation Biology, Institute of Zoology, Chinese Academy of Sciences, Beijing 100101, China; 3School of Karst Sciences, Guizhou Normal University, Guiyang 550001, China; 4Administration of Daxiangling Nature Reserve, Yaan 625000, China

**Keywords:** *Ailuropoda melanoleuca*, Niba Mountain corridor, suitable habitat, habitat selection, principal component analysis

## Abstract

**Simple Summary:**

The giant panda is an endemic species in China and the flagship species of global wildlife conservation. Habitat studies of the giant panda corridor can reveal their survival status, habitat environment, and the threats they face, which is crucial for giant panda population recovery and habitat conservation. The results of this study show that due to the opening of the National Road 5 (G5) Niba Mountain tunnel and the completion of the Niba Mountain giant panda corridor plan, the recovery of vegetation within the Niba Mountain giant panda corridor has led to the emergence of giant panda activity in the area, which may have spread to the central part of the reserve through the corridor. The habitat selection characteristics of the giant pandas in the corridor were clarified by investigating the microhabitats of the giant panda corridor in Niba Mountain. These findings can provide a reference for scientists to formulate practical habitat conservation and management measures for giant pandas in the study area.

**Abstract:**

Habitat reduction and increased fragmentation are urgent issues for the survival and recovery of the giant panda (*Ailuropoda melanoleuca*). However, changes in the distribution and microhabitat selection of giant panda habitats in different seasons in the same region have rarely been assessed. To further understand giant panda habitat requirements, this study analyzed the giant panda habitat selection characteristics and differences using the sample data of the giant panda occurrence sites collected during 2020–2022. The results showed that the giant panda in both seasons selected medium altitudes (2000–2400 m), southeastern slopes, slopes less than 15°, taller tree layers (8–15 m) with a larger diameter at breast height (17–25 cm) and medium density (25–55%), shorter shrub layers (<4 m) with sparse density (<30%), and taller bamboo (>2 m) with high density (>35%). The giant panda microhabitat survey in the Niba Mountain corridor clarified the characteristics of suitable habitat selection for the giant panda in the corridor. The findings of the study can provide scientific references for the development of practical habitat conservation and management measures for giant pandas in the study area.

## 1. Introduction

The giant panda is an endemic species in China and the flagship species of global wildlife conservation. Currently, the species is distributed in six mountains: Qinling [1], Minshan [2], Qionglai [3], Daxiangling [4], Xiaoxiangling [5], and Liangshan [6,7]. In recent decades, the giant panda population has decreased, with suitable habitat areas becoming increasingly shrunken and fragmented due to an imbalance between economic development and ecological conservation [8,9]. Habitat reduction and increased fragmentation are urgent challenges for the survival and recovery of the giant panda today [10].

To effectively protect the giant panda’s habitats, the establishment of habitat corridors for the giant panda has been widely considered [11]. In 2007, with the support of the World Wildlife Fund, the earliest exploration of giant panda habitat corridor construction began in the Tudeling area, connecting the A and B populations of the giant panda in Minshan [12]. According to the Fourth National Giant Panda Survey (2011–2014), four giant panda corridor zones were identified in the distribution area of the giant panda. The Niba Mountain giant panda corridor zone, located in the Daxiangling Mountains, was identified as a priority ecological corridor for construction [13]. There has been no historical distribution of the giant panda in Niba Mountain, and no traces of giant panda activity were found in the area during the Fourth National Giant Panda Survey. Niba Mountain was classified as a key corridor area connecting the Daxiangling giant panda population and the Qionglai population. The construction of giant panda corridors is important for alleviating the declining genetic diversity of giant panda populations among different mountains [14]. In recent years, Gong et al. found that the habitat pattern of the giant panda is an important basis for corridor site selection. Additionally, they suggested that the study of all habitat-related microhabitat factors should be focused on [15]. Habitat studies of the giant panda corridor can more fully reveal the giant panda’s survival status, habitat environment, and threats they face. This is crucial for the population recovery and habitat conservation of wild giant pandas [16].

Various constraints affect the habitat selection of the giant panda, including topography (elevation, slope, and aspect) and community structure (tree size and bamboo cover) [17]. These constraints may affect giant panda mobility, shelter availability, and the palatability of edible bamboo for the panda [18]. Previous studies showed that between 1999–2000 and 2011–2014, the giant panda in several mountains in Sichuan experienced a shift in habitat use. The giant panda increasingly utilized secondary forests, which had recovered due to protective measures. The giant panda migrated to higher elevations, despite the availability of bamboo food sources at lower elevations [10]. The implementation of natural forest conservation programs, infrastructure construction, livestock encroachment, and a range of emerging threats may have affected giant panda habitat selection [10].

Previous studies have also investigated the giant panda’s habitat selection in the Daxiangling Mountains and reported changes in the habitat selection of this species between 2001 and 2020 [4,19,20]. However, these studies only described the overall habitat selection of the giant panda throughout the year. The habitat selection and utilization of the giant panda under the influence of different environmental conditions in different seasons were not further investigated. The habitat selection of the giant panda varies over time. Furthermore, they found that highly edible bamboo and good shelter sites had an important influence on the habitat selection of the giant panda [18]. The giant pandas feed mainly on *Chimonobambusa szechuanensis* and *Qiongzhuea multigemmia* at lower elevations (1500–2200 m) and *Arundinaria faberi* at higher elevations (2000–2600 m) in the Daxiangling Mountains [21,22]. However, changes in the distribution and microhabitat selection of giant panda habitats in different seasons in the same region have rarely been assessed. Determining a suitable habitat distribution pattern and selection characteristics within the protected area is essential for formulating a scientific and reasonable giant panda conservation and management plan [8]. Therefore, based on the sample data of giant panda occurrence sites collected during 2020–2022, this study aimed to apply MaxEnt and other methods to evaluate the habitat selection characteristics, including the spatial distribution pattern of suitable habitats for the giant panda in the Daxiangling Niba Mountain corridor [23]. The research conclusions can provide a scientific reference for giant panda habitat conservation and management in the area.

## 2. Materials and Methods

### 2.1. Study Area

The Daxiangling Mountains are located in the transition zone between the Sichuan Basin and the Qinghai–Tibet Plateau in the eastern part of the Hengduan Mountains, covering an area of about 6440 km^2^. The vegetation in the region has obvious vertical distribution zones, and the distribution order is evergreen broad-leaved forest (below 1400 m), deciduous broadleaf forest (1400–1800 m), mixed coniferous forest (1800–2600 m), coniferous forest (2600–3100 m), and alpine scrub meadow (above 3100 m) [24].

According to the results of the Fourth National Giant Panda Survey, there are 14 wild giant pandas distributed in the Daxiangling Reserve and Longcang Valley [18]. Daxiangling is the main conservation area of the Yingjing Area of Giant Panda National Park and one of the key areas of Giant Panda National Park [25,26]. The Daxiangling Niba Mountain giant panda ecological corridor (102°29′–102°52′ E, 29°28′–29°43′ N) is located in the southwestern part of the reserve in Yingjing County and Hanyuan County, Sichuan Province, covering an area of about 38.5 km^2^. The National Road 108 (G108) Ya’an section and the G5 Yaxi Expressway Niba Mountain tunnel were completed and opened to traffic in 2000 and 2012, respectively. Both cross the corridor from north to south. For the construction of these highways, many bamboo trees were cut down, which are major food sources for pandas. Furthermore, frequent human activities have caused the separation of the giant panda habitat in Daxiangling [24]. The Niba Mountain corridor has also become a key area for the exchange of giant panda populations in the Daxiangling Mountains [13].

### 2.2. Giant Panda Habitat Selection Analysis

From October 2020 to April 2022, the data of giant panda traces were recorded using the sample line method and infrared camera monitoring method, and a sample survey was conducted in March–April and October–November (from 2020 to 2022) in the areas where giant panda traces were recorded to determine the preferred environmental factors of the giant panda habitat. A total of 34 sample lines ≥3 km were set at intervals of ≥500 m. The sample lines covered as many vegetation types and as many potential giant panda distribution areas as possible. Combining data from 158 infrared cameras placed in the study area, the entire Daxiangling Reserve was divided into 145 square grids of 2 km^2^ each, with each camera spaced at least 500 m apart to ensure uniform camera coverage (Figure 1). Ten microhabitat variables were recorded in a 10 × 10 m sample square centered on the site of giant panda traces. The classification criteria for different environmental variables are shown in Table 1. A control sample was randomly set up along the sample line for every 500 m of walking or 100 m of elevation climb without traces of giant panda activity to reflect the environmental background information, and the setting and habitat variables of the control sample were recorded in the same way as the utilization sample [27]. A total of 348 samples were set up [23].

These habitat selection and ecological niche data were input into Excel for the relevant conversions. Following the conversion, the data were entered into SPSS13.0 for normality testing via the one-sample K-S test. Data that conformed to a normal distribution were tested through one-way analysis of variance (ANOVA), and data that did not conform to a normal distribution were tested using the Mann–Whitney *U* test.

### 2.3. Construction of a Suitable Habitat Model for the Giant Panda in the Niba Mountain Corridor

The estimation of suitable habitats for giant pandas in the study area was performed using the MaxEnt model. A total of 62 giant panda occurrence sites were obtained in the field, with 44 and 18 occurrence sites in the rainy and snow seasons, respectively. To reduce autocorrelation, an 1125 m radius buffer was generated in ArcGIS 10.2 with giant panda occurrence sites in the rainy and snow seasons. When the occurrence site buffers overlapped with each other, one of them was randomly retained, and the rest were eliminated, resulting in 20 and 10 occurrence sites retained in the rainy and snow seasons, respectively [27]. The giant panda has rigorous requirements for habitat, usually choosing primary forests with low human interference [28,29]. Climate and land-use types are also important factors that influence the spatial distribution of the giant panda [9,30]. Therefore, climate, topography, vegetation, and human disturbance are important factors affecting the spatial distribution of the giant panda. In the construction of the model, the above variables were selected to evaluate the habitat. Access to each variable is shown in Table 2 [23]. Because the prediction accuracy of the model was affected by the correlation between environmental factors, the “caret” function in R 4.2.1 was used to remove the highly correlated variables, and the factors with Pearson correlation coefficients greater than 0.8 were removed. Finally, nine factors were retained (Table 2).

In total, 75% of the occurrence sites of the giant panda were selected for modeling, and 25% of the occurrence sites were retained for validation. The importance of each environmental factor was assessed using the Jackknife method, and the output was in the logistic format. The model prediction results were tested using the receiver operating characteristic (ROC) curve [31]. The evaluation criteria were as follows: the area enclosed by the ROC curve and the area under the curve (AUC value) was 0.5–0.6 for failure, 0.6–0.7 for poor, 0.7–0.8 for fair, 0.8–0.9 for good, and 0.9–1.0 for excellent [32]. The means of 10 calculation results were averaged to gain the habitat suitability index (HSI). The suitable habitat range in the study area was divided using Youden’s index as the threshold.

### 2.4. PCA of Microhabitat Factors for Giant Panda Habitat Selection in the Niba Mountain Corridor

PCA can project the high-dimensional original data onto the low-dimensional mutually orthogonal principal components. This process can maximize the information content of the original data while reducing the dimensionality of the data and can effectively overcome the correlation or multicollinearity problem between variables through mutually orthogonal principal components [27]. The raw data of giant panda microhabitat variables were analyzed using PCA; the mean and covariance matrices of the sample data matrix were calculated, and the eigenvalue of the correlation matrix was found. The principal components with eigenvalues >1 were extracted, and each principal component and its contribution rate were derived from the eigenvalue to determine the factors that play a major role in giant panda habitat selection [33]. PCA was performed in IBM SPSS Statistics 26.0. In the statistical analysis, *p* < 0.05 was considered statistically significant.

## 3. Results

### 3.1. Results of Giant Panda Habitat Selection Analysis

The giant panda occurrence sites included 18 sites recorded using the sample line method and 26 sites photographed by infrared cameras in the rainy season. All occurrence sites were photographed by infrared cameras during the snow season. The total number of control samples was 155 in the rainy season and 131 in the snow season (Table 3).

The one-sample K–S test was performed for 10 ecological factors in the habitat selection and control plots. The results showed that six variables—namely, the altitude, average height of trees, diameter at breast height of trees, tree coverage, the average height of shrub, and bamboo height—were normally distributed, while the other four variables, including aspect, were not. The plot type was used as the control variable for the ANOVA. In normally distributed variables, there were significant differences between altitude, the average height of trees, and diameter at the breast height of trees (*p* < 0.05), but no significant differences between tree coverage, average height of trees, and the average height of bamboo (*p* > 0.05). In the control and habitat selection plots, the Mann–Whitney *U* test revealed significant differences between two variables, namely shrub cover and bamboo cover (*p* < 0.05). However, there was no significant difference between the aspect and slope (*p* > 0.05; Table 4).

In both seasons, giant pandas preferred medium altitudes (2000–2400 m), southeastern slopes, and areas with slopes of 15°. The average elevation preferred by the giant panda in the snow season (2209 m) was higher than that in the rainy season (2147 m), i.e., a difference of 62 m. The giant panda was captured at an average aspect of 158.98° during the rainy season and 141.39° during the snow season. Furthermore, the giant panda was captured at an average slope of 13.41° during the rainy season and 8.89° during the snow season.

The preferred community structure of the giant panda habitat was characterized by a preference for tall (8–15 m), large (17–25 cm) diameter at breast height, and moderately dense (25–55%) trees; short (<4 m) and sparse (<30%) shrub; and tall (>2 m) and dense (>35%) bamboo trees in both seasons. The mean height of trees during the rainy season (10.98 m) and the snow season (12.28 m) were both greater than 8 m. The mean diameter at the breast height of trees during the rainy season (24.27 cm) and the snow season (17.72 cm) were both greater than 17 cm. Furthermore, the depression of trees during the rainy season (38.30%), and the snow season (40.56%), were both greater than 25%. The mean height of the shrub during the rainy season (2.47 m) and the snow season (3.58 m) were both less than 4 m. Additionally, the shrub cover during the rainy season (18.75%) and the snow season (27.22%) were both less than 30%. The mean height of the bamboo during the rainy season (2.12 m) and the snow season (2.39 m) were both greater than 2 m. Furthermore, the bamboo cover during the rainy season (59.32%) and the snow season (63.06%) were both greater than 30%.

### 3.2. Assessment of Suitable Giant Panda Habitat in the Niba Mountain Corridor

According to the ROC test results of the Maxent model, the AUC values were 0.964 and 0.967 for the rainy and the snow seasons, respectively, the prediction accuracy reached the level of “excellent,” and the maximum Youden’s index values were 0.316 and 0.527, respectively. The prediction results were transformed in ArcGIS 10.2. The results showed that the suitable habitat areas for the giant panda in the rainy and snow seasons were about 95.61 km^2^ and 41.56 km^2^, respectively, accounting for 7.30% and 3.17% of the total area of the study area, respectively. In addition, most of the suitable habitat areas for the giant panda in the rainy and snow seasons were located between 1800–2600 m (mixed coniferous and broad forest). The area of suitable habitat for the giant panda in the snow season overlapped with that in the rainy season, while the overlapping area of suitable habitat in the rainy and snow seasons was about 26.85 km^2^, with an area overlap rate of 24.34%. Suitable habitats in the rainy season were located in the western part of the Daxiangling Reserve, in Niba Mountain, and the Longcang Valley (Figure 2), while suitable habitats in the snow season were mainly located in Niba Mountain (Figure 3). The area of the west side of G108 in the rainy and snow seasons was about 55.88 km^2^ and 19.86 km^2^, respectively, while the area of the east side of G108 was approximately 22.23 km^2^ and 14.12 km^2^, respectively. The suitable habitat during the rainy season was located in the eastern periphery of the Daxiangling Reserve in the Longcang Valley, with an area of about 17.14 km^2^; the suitable habitat during the snow season was also scattered in the eastern part of the Reserve, and the periphery of the Longcang Valley, with an area of about 7.07 km^2^. The distribution of suitable habitats in both the rainy and snow seasons showed obvious fragmented characteristics.

### 3.3. Main Factors Affecting Microhabitat Selection in the Giant Panda

Among the 10 principal components of the rainy season and snow season, the eigenvalues of the first four principal components were >1, and the cumulative contributions reached 62.71% and 58.64% during the rainy and snow seasons, respectively. The first four principal components were extracted, and their corresponding eigenvectors were calculated. The variables with higher loadings in PC1 during the rainy and snow seasons were the mean height of the tree layer (10.98 m), the mean height of bamboo (2.12 m), the mean height of the tree layer (12.28 m), and the mean diameter at breast height of the tree layer (17.72%). The variables with higher loadings in PC2 were the mean height of the shrub layer (2.47 m), the shrub layer cover (18.75%), the mean height of bamboo (2.39 m), and the bamboo cover (63.06%). The variables with higher loadings in PC3 were the mean height of bamboo (59.32%) and the shrub layer cover (27.22%). Furthermore, the variables with higher loadings in PC4 for both the rainy and snow seasons were altitudes of 2147.84 m and 2209.17 m, respectively (Table 5 and Table 6).

## 4. Discussion

### 4.1. Main Activity Area of the Giant panda in the Niba Mountain Corridor

According to the data from previous giant panda surveys, there were no traces of giant panda activity in Niba Mountain before 2015 [18]. A trail of giant panda activity was discovered in Niba Mountain in 2016. Infrared cameras began to be placed in 2017, and giant panda activity was photographed in 2018. The traces of giant panda activity were recorded during different seasons in the Niba Mountain corridor (Figure 1). Furthermore, the giant pandas were spread out in the middle of the corridor. Therefore, the giant panda may have been able to use the corridor to migrate, and the planning of the giant panda corridor in Niba Mountain is reasonable.

In 2012, the G5 Niba Mountain tunnel was completed and put into use, and the traffic flow of the G108 dropped sharply, which greatly reduced the transportation disruption in Niba Mountain. However, the vegetation in the area is still in a slow recovery stage, and no giant panda activity was found during the Fourth Survey [34]. In 2015, the planning of the Niba Mountain giant panda corridor was completed, and vegetation restoration work began in Niba Mountain [13]. Traces of giant panda activity were recorded in 2016, which indicates that it will take at least 4 years for the vegetation within an 1125 m radius of giant panda activity on the left and right sides of G108 to recover to a stage that can be utilized by the giant panda. We found that some human activities, such as bamboo shoot collection, grazing, and medicinal plant collection, occurred within an 1125 m radius of giant panda activity. However, human activities are controlled on a limited scale. Therefore, no major disturbance impacting the activities of the giant panda in the area has occurred [35].

### 4.2. Giant Panda Microhabitat Characteristics in the Niba Mountain Corridor

Our study showed that there were distinctive features of giant panda habitat selection in the study area. The giant panda chose to move in southeastern, gently sloping areas at medium altitudes with tall trees in both seasons, which is consistent with the results reported by Fu et al. and Bai et al. [17,18]. The giant panda always chose habitats with lower energy consumption requirements as well as higher nutritional value and net energy gain [36]. A survey of the microhabitats of the giant panda in Niba Mountain clarified that the suitable habitat characteristics of the giant panda in this area were similar to those in other areas of Sichuan Province [17,23,37].

The giant panda preferred to move at moderate altitudes in both seasons, but the average distribution elevation in the snow season was about 62 m higher than that in the rainy season. We believe that this may be related to their preference to feed on bamboo species in different seasons. The giant panda prefers to feed on *Qiongzhuea multigemmia* and *Chimonobambusa szechuanensis* at relatively low altitudes (1500–2200 m) in the rainy season and *Arundinaria faberi* at relatively high altitudes (2000–2600 m) in the snow season. This indicates that the giant panda constantly migrates according to the seasons to meet their energy needs [38], similar to the findings of Chen et al. and Liu et al. on the existence of “seasonal vertical movement” in the giant panda [39,40]. Human activities in the Daxiangling Mountains are mainly concentrated in valley areas at low altitudes (<1400 m). There are few human activities in the high-altitude areas (>1800 m) where the giant panda is active, and the giant panda can avoid anthropogenic disturbance by choosing middle- and high-altitude areas as their habitat [10,11]. The results of PCA showed that altitude and bamboo growth status were significantly associated with the habitat preference of the giant panda. This is because different altitudes can provide different cumulative temperatures for the bamboo. For example, *Qiongzhuea multigemmia* only reaches the conditions for shoot development at an altitude of 1500–2200 m above sea level and when its accumulated temperature reaches a certain standard, coupled with the high ambient precipitation in March [41]. Our results indicated that altitude affects the bamboo shoot collection, the staple food of the giant panda, and thus influences the habitat selection of the giant panda in different seasons.

The Maxent model was used to predict the exact distribution and area of suitable giant panda habitats in the region during different seasons. The distribution of suitable giant panda habitats was more fragmented in snow than in rainy seasons. With the construction of the corridor, the suitable habitat area for the giant panda in the Daxiangling Reserve and Longcang Valley reached about 95.61 km^2^ in the rainy season and 41.56 km^2^ in the snow season, which can promote further rejuvenation of the giant panda population. The suitable habitat area for giant pandas in the snow season overlapped with that during the rainy season. The overlapping area was about 26.85 km^2^, which accounts for 24.34% of the total suitable habitat area. The overlapping area, which was primarily distributed in Niba Mountain, was also the core area for the giant panda’s year-round activities [40]. This is because the giant panda prefers to select different bamboo species distributed at different elevations and in different seasons. Furthermore, this verifies the research result that the average distribution elevation of the giant panda in the snow season is about 62 m higher than in the rainy season. The next step will be to introduce a finer level (four levels: most suitable, suitable, less suitable, and unsuitable) of assessment criteria for giant panda habitat assessment in the study area to explore the potential habitat and dispersal pathways of giant pandas in the study area.

The results showed significant seasonal differences in the selection of microhabitats by the giant panda, reflecting that the species has different resource requirements in different seasons [42].

### 4.3. Newly Recorded Site for Giant Panda Activity in the Middle of the Niba Mountain Corridor

A giant panda activity trail was discovered in the central part of the Niba Mountain corridor in 2019. Two infrared cameras captured giant panda activity in February 2022. Both sites are located within the Niba Mountain giant panda corridor, and it is possible that the giant panda in Niba Mountain is spreading westward along the corridor or the giant panda in the Longcang Valley is spreading eastward. The Niba Mountain corridor is a dispersal channel for the giant panda populations on the left and right sides, creating conditions for the exchange of giant panda populations between the two sides. However, at present, the Niba Mountain corridor is only planned to be about 5 km east of the G5 in the central part of the reserve and does not reach the Longcang Valley. Therefore, it is suggested that another ecological corridor for the giant panda can be planned from the central part of the reserve to the Longcang Valley. Alternatively, the eastern part of the Niba Mountain corridor can be extended to the Longcang Valley to strengthen the exchange of giant panda populations [18]. In 2016, a new point of giant panda activity was discovered in Niba Mountain, and in 2019, it was discovered in the middle of the Niba Mountain corridor. This indicates that it takes about 3 years for the giant panda to move from the west to the east of the reserve and from the north to the south [35].

We are aiming to better verify the actual role of the Niba Mountain giant panda corridor in the dispersal of the giant panda. In the next step, biological samples such as fresh feces will be collected from the giant panda in different areas. Then, we will explore the genetic diversity of the giant panda population by combining microsatellite and mitochondrial control regions, using feces as the main experimental material. Furthermore, we will explore the origin of newly recorded giant panda occurrence sites in the central part of the country to determine the dispersal path of the giant panda population in the Daxiangling Mountains [43].

As one of the important methods used to connect local populations of the giant panda and restore their habitat, giant panda corridors have been established in key areas under the requirements of China’s giant panda habitat management plan. The establishment of these corridors has restored vegetation in the relevant areas, connected the more severely fragmented giant panda populations, and promoted exchanges between various giant panda populations [44]. The construction of the G5 Niba Mountain tunnel also provides a reference for the conservation of giant panda habitat in other mountain systems in Sichuan Province, demonstrating that it is possible to minimize the impact on the connectivity of the original giant panda habitat using elevated roadways or tunnels. This strategy not only meets the transportation development needs of human society but also maintains the normal distribution and dispersal of giant panda populations to a certain extent. We recommend monitoring the movement of the giant panda near roads and their use of corridors (e.g., road tunnels) to evaluate the impact of corridors on the giant panda. Further research should assess the effects of bamboo cover on the foraging and movement of the giant panda. In addition, it is suggested that field patrols be strengthened to decrease human activities in the giant panda habitat, including grazing, herb collection, bamboo shoot collection, and hunting. The future production and living modes of residents can be driven by ecological tourism development and the exploration of agricultural and sideline products with regional characteristics, which may change the negative attitudes of residents toward ecological conservation. This method can promote the industrial transformation to realize the balanced development of economic construction and ecological conservation [18]. Additionally, the Daxiangling Mountains should focus on the continuous conservation of major habitats for the giant panda, decrease habitat fragmentation caused by anthropogenic interference, and strengthen habitat restoration for the giant panda [23].

## 5. Conclusions

The present study shows that the giant panda prefers to move to medium-altitude areas; southeastern slopes; and areas with slopes less than 15°, with an overlap of suitable habitats in both seasons. Due to the opening of the G5 Niba Mountain tunnel and the completion of the Niba Mountain giant panda corridor plan, the recovery of vegetation within the Niba Mountain giant panda corridor has led to the emergence of giant panda activity in the area, which may have spread to the central part of the reserve through the corridor. The findings can provide a reference for the future planning and construction of giant panda corridors.

## Figures and Tables

**Figure 1 biology-12-00165-f001:**
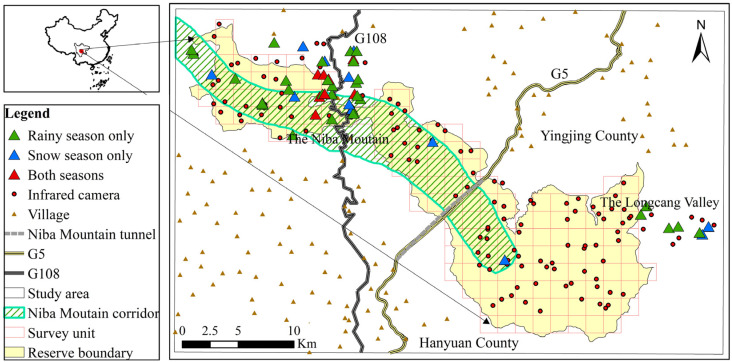
Map of infrared camera sites and giant panda occurrence sites in the study area.

**Figure 2 biology-12-00165-f002:**
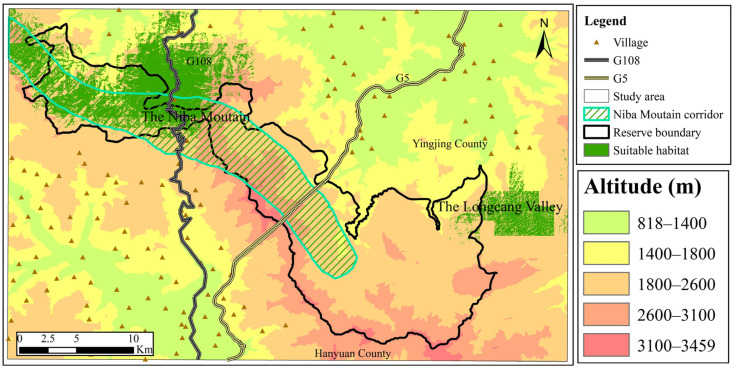
Suitable giant panda habitat during the rainy season.

**Figure 3 biology-12-00165-f003:**
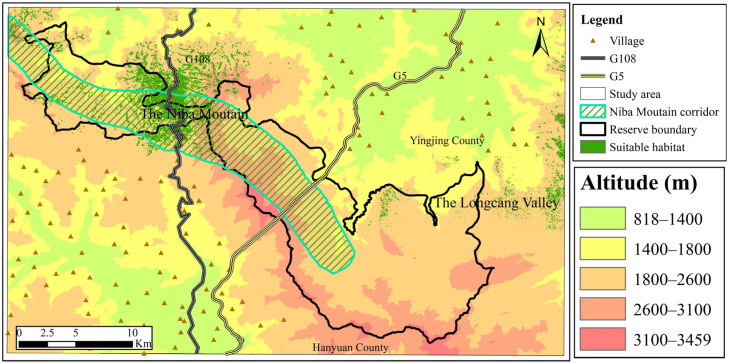
Suitable giant panda habitat during the snow season.

**Table 1 biology-12-00165-t001:** Classification criteria for microhabitat variables.

Microhabitat Variable	Classification Criteria
Altitude	<1400 m (evergreen broadleaf forests); 1400–1800 m (deciduous broadleaf forests); 1800–2600 m (mixed coniferous forests); 2600–3100 m (coniferous forests); >3100 m (alpine scrub meadows)
Aspect	<22.5°, >337.5°(N); 22.5°–67.5°(NE); 67.5°–112.5°(E); 112.5°–157.5°(SE); 157.5°–202.5°(S); 202.5°–247.5°(SW); 247.5°–292.5°(W); 292.5°–337.5°(NW)
Slope	<15°; 15°–30°; >30°
Average height of trees	<8 m; 8–12 m; >12 m
Average diameter at breast height of trees	<17 cm; 17–25 cm; >25 cm
Tree coverage	<25%; 25–55%; >55%
Shrub height	<3 m; 3–4 m; >4 m
Shrub coverage	<30%; 30–50%; >50%
Average height of bamboo	<1 m; 1–2 m; >2 m
Bamboo coverage	<20%; 20–35%; >35%

**Table 2 biology-12-00165-t002:** List of environmental factors for habitat suitability evaluation of the giant panda in the study area.

Types	Factor Codes	Description of Factors	Unit	Data Sources
Climate	Bio2	Mean Diurnal Range	°C	Worldclim (http://www.worldclim.org/, accessed on 7 November 2021)
	Bio7	Temperature Annual Range	°C	
	Bio11	Mean Temperature of Coldest Quarter	°C	
Topography	Aspect	Aspect	°	Geospatial data cloud (http://www.gscloud.cn/, accessed on 6 November 2021)
	Slope	Slope	°	
Interfere	Road	Distance to roads	m	The 4th Survey Report on Giant Panda in Sichuan Province
	Village	Distance to villages	m	
Resources	River	Distance to rivers	m	Resource and environment science and data center (http://www.resdc.cn/, accessed on 6 November 2021)
	Vegetation	Vegetation types categorical variable, divided into six categories: evergreen broadleaf forest, deciduous broadleaf forest, mixed coniferous forest, coniferous forest, alpine scrub meadows, and other lands	/	

**Table 3 biology-12-00165-t003:** Comparison of microhabitat characteristics of the giant panda during the rainy and snow seasons (mean ± standard deviation).

Microhabitat Variable	Rainy Season			Snow Season		
	Microhabitat	Control	Advantageous Plants	Microhabitat	Control	Advantageous Plants
Altitude (m)	2147.84 ± 232.44	2153.98 ± 386.02		2209.17 ± 227.09	2074.75 ± 294.52	
Aspect (°)	158.98 ± 96.05	184.65 ± 106.14		141.39 ± 115.69	177.97 ± 104.63	
Slope (°)	13.41 ± 9.32	14.41 ± 10.18		8.89 ± 5.87	12.93 ± 10.29	
Average height of trees (m)	10.98 ± 4.60	12.61 ± 5.36	*Acer oliverianum Pax* (9.41 ± 4.17)	12.28 ± 3.88	11.83 ± 4.22	*Abies fabri* (13.27 ± 3.80)
Average diameter at breast height of trees (cm)	24.27 ± 14.32	19.23 ± 8.80	27.41 ± 19.46	17.72 ± 6.06	19.88 ± 8.78	19.27 ± 6.23
Tree coverage (%)	38.30 ± 16.21	40.14 ± 18.16	37.94 ± 18.63	40.56 ± 16.35	38.00 ± 14.64	42.73 ± 11.26
Average height of shrub (m)	2.47 ± 1.60	3.21 ± 1.19		3.58 ± 0.58	3.34 ± 0.87	
Shrub coverage (%)	18.75 ± 16.54	26.84 ± 16.36		27.22 ±7.71	24.42 ± 10.85	
Average height of bamboo (m)	2.12 ± 1.11	1.77 ± 1.45	*Qiongzhuea multigemmia* (1.91 ± 0.47)	2.39 ± 0.54	1.78 ± 1.37	*Arundinaria faberi* (2.36 ± 0.46)
Bamboo coverage (%)	59.32 ± 21.20	41.87 ± 31.71	57.78 ± 21.85	63.06 ± 24.32	35.00 ± 29.51	64.50 ± 23.51

**Table 4 biology-12-00165-t004:** Comparison of variables in habitat selection plots and control plots in the study area by analysis of variance (ANOVA) and Mann–Whitney *U* tests.

Microhabitat Variable	Rainy Season		Snow Season	
	ANOVA	Mann–Whitney *U* Test	ANOVA	Mann–Whitney *U* Test
	*F* (*p*)	*U* (*p*)	*F* (*p*)	*U* (*p*)
Altitude (m)	40.04 (0.00 ***)		36.07 (0.00 ***)	
Aspect (°)		171,173 (0.17)		165,673 (0.08)
Slope (°)		167,424 (0.10)		197,344 (0.11)
Average height of trees (m)	35.70 (0.00 ***)		45.50 (0.00 ***)	
Average diameter at breast height of trees (cm)	9.95 (0.01 **)		10.03 (0.01 **)	
Tree coverage (%)	2.76 (0.97)		2.67(0.93)	
Average height of shrub (m)	0.15 (0.70)		0.21 (0.83)	
Shrub coverage (%)		163,905 (0.04 *)		164,335 (0.04 *)
Average height of bamboo (m)	6.20 (0.13)		6.17 (0.11)	
Bamboo coverage (%)		146,994 (0.00 ***)		133,885 (0.00 ***)

* *p* < 0.05, ** *p* < 0.01, *** *p* < 0.001.

**Table 5 biology-12-00165-t005:** Principal component analysis loadings of habitat factors at the microhabitat scale of the giant panda during the rainy season.

Microhabitat Variable	Rainy Season			
	PC1	PC2	PC3	PC4
Altitude (m)	−0.023	0.000	−0.400	0.516
Aspect (°)	0.074	0.170	0.031	−0.781
Slope (°)	−0.244	0.176	−0.302	0.148
Average height of trees (m)	0.711	0.413	−0.254	−0.007
Average diameter at breast height of trees (cm)	0.679	0.129	−0.307	−0.176
Tree coverage (%)	0.567	0.277	−0.367	0.059
Average height of shrub (m)	0.065	0.790	0.387	0.203
Shrub coverage (%)	−0.267	0.776	0.347	0.055
Average height of bamboo (m)	0.714	−0.229	0.372	0.165
Bamboo coverage (%)	0.552	−0.343	0.569	0.145
Eigenvalue	2.245	1.721	1.279	1.026
Contribution (%)	22.450	17.213	12.790	10.258
Cumulative contribution (%)	22.450	39.663	52.453	62.711

**Table 6 biology-12-00165-t006:** Principal component analysis loadings of habitat factors at the microhabitat scale of the giant panda during the snowy season.

Microhabitat Variable	Snow Season			
	PC1	PC2	PC3	PC4
Altitude (m)	−0.154	0.257	−0.242	0.616
Aspect (°)	−0.235	−0.018	0.204	−0.672
Slope (°)	−0.038	−0.231	−0.324	0.360
Average height of trees (m)	0.810	−0.051	0.206	0.062
Average diameter at breast height of trees (cm)	0.785	−0.109	0.000	−0.055
Tree coverage (%)	0.642	−0.170	0.098	0.084
Average of shrub height (m)	−0.083	0.365	0.604	0.274
Shrub coverage (%)	−0.136	0.227	0.706	0.216
Average height of bamboo (m)	0.252	0.802	−0.160	−0.149
Bamboo coverage (%)	0.153	0.792	−0.295	−0.133
Eigenvalue	1.877	1.617	1.233	1.136
Contribution (%)	18.772	16.175	12.330	11.359
Cumulative contribution (%)	18.772	34.947	47.277	58.636

## Data Availability

Data are available on request.

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
