# Peer review of "Giant Panda Microhabitat Study in the Daxiangling Niba Mountain Corridor"

_biology, 2023, doi:10.3390/biology12020165_

Round 1
Reviewer 1 Report
On an overall basis, the work is of strong scientific significance. The study clarifies the microhabitat selection characteristics of giant pandas in the corridor through microhabitat surveys of giant pandas in the Niba Mountain corridor , which has certain guiding significance for the rejuvenation of giant panda populations. but, the overall presentation of the article is not concise and compact, and there is room for further improvement of the logical structure, and we hope the authors will further improve it according to the revision comments. Specific details of the revisions are as follows.
1. It is necessary to add the introduction of the historical background of Niba Mountain in the introduction section, introducing whether there were giant pandas in the history of the Niba Mountain area, and if so before, when there were no giant pandas.
2. The distance of anthropogenic disturbance activities from giant panda activity locations needs to be clarified, whether they are all outside the buffer zone of 1125m radius of giant panda activities; whether the intensity of anthropogenic disturbance activities are all below 1800m above altitude of giant panda activities.
3. Suggest deleting the repetition of the discussion section and the results section, and giving the main reasons for the difference in altitude between the rainy and snowy seasons of giant pandas in the discussion.
4. Principal component analysis has already analyzed the habitat features, and the logistic Stiff model part of the R language is analyzed again, which in my opinion duplicates the work.
Author Response
Response to the First Reviewer:Dear Reviewer,Thank you for your comments on our manuscript title " Giant Panda Microhabitat Study in the Daxiangling Niba Mountain Corridor (ID: biology-2123287)". Your comments have been helpful in revising and improving our manuscript and will be an important guide for our future research. We have carefully studied these comments and have worked hard to revise the manuscript in the hope of obtaining your approval. The revisions are marked in red in the manuscript. The following is a response to your comments:
Comment 1: It is necessary to add the introduction of the historical background of Niba Mountain in the introduction section, introducing whether there were giant pandas in the history of the Niba Mountain area, and if so before, when there were no giant pandas.
Response: Dear reviewer, based on your comments, we have added the introduction of the historical background of Niba Mountain in the introduction section. You can see the revised content below.There has been no historical distribution of giant pandas in Niba Mountain, and no traces of giant panda activity were found in Niba Mountain during the Fourth National Giant Panda Survey. Niba Mountain was classified as a key corridor area connecting the Daxiangling giant panda population and the Qionglai population.
Comment 2: The distance of anthropogenic disturbance activities from giant panda activity locations needs to be clarified, whether they are all outside the buffer zone of 1125m radius of giant panda activities; whether the intensity of anthropogenic disturbance activities are all below 1800m above altitude of giant panda activities.
Response: Dear reviewer, based on your comments, we have added that some human activities occur within the 1125m buffer zone of the giant panda activity radius and that there are few human disturbance activities in the high altitude (>1800m) areas of the giant panda activity. You can see the revised content below. Although we found that some human activities occurred within the buffer zone of 1125m of the activity radius of giant pandas, the human disturbance activities in the area such as bamboo shooting, grazing, and medicine collection are controlled to a limited scale, and therefore no major disturbance impact on the activities of giant pandas in the area has occurred. Human activities in the Daxiangling area are mainly concentrated in the valley areas at low altitudes (<1400 m), there are few human disturbance activities in the high altitude (>1800m) areas where giant pandas are active, and giant pandas can avoid anthropogenic disturbance by choosing middle and high-altitude areas as their habitat.
Comment 3: Suggest deleting the repetition of the discussion section and the results section, and giving the main reasons for the difference in altitude between the rainy and snowy seasons of giant pandas in the discussion.
Response: Thank you. Based on your comments, we have removed the duplication in the results and discussion and have given the main reasons for the difference in elevation between the rainy and snow seasons of giant pandas in the discussion. You can see the revised content below.
We believe that the main reason for the difference in elevation between the rainy and snow seasons of giant pandas is the difference in staple bamboo species, with giant pandas prefer to use Qiongzhuea multigemmia and Chimonobambusa szechuanensis at relatively low altitudes (1500–2200 m) in the rainy season and Arundinaria faberi at relatively high altitudes (2000–2600 m) in the snow season.
Comment 4: Principal component analysis has already analyzed the habitat features, and the logistic Stiff model part of the R language is analyzed again, which in my opinion duplicates the work.
Response: Thank you. Based on your comments, we have removed the logistic Stiff model part of the R language as it appeared to duplicate the results of the principal component analysis.

Reviewer 2 Report
This study is clear and meaningful, and the data of this study are solid. However, the manuscript and methods have some problems. I list my comments below:
Line 24: It is recommended that this sentence be deleted from the Abstract.
Line 37-39: Giant pandas are now distributed in six mountain ranges, and the source of references indexed by them is not representative enough or misquoted. It is suggested to revise it to the latest national census report of giant pandas.
Line 59-60: At least representative articles on giant panda microhabitat selection should be cited.
Line 62-63: It is suggested to point out how the habitat use changes and mark this reference.
Line 72-28: The logicality of this paragraph needs to be improved. For example, what is the purpose of evaluating habitat suitability?
Line 107-112: The time of the sample survey should be clearly stated.
Table 2: There is no elevation factor, which seems difficult to understand.
Line 197-199: The AICc value of the model included in the average can be directly expressed, not just a reference.
3.1. Results of Giant Panda Habitat Selection Analysis: At least a simple difference test should be added to this part to analyze the seasonal difference of giant panda habitat selection in the study area. The current descriptive statistics cannot reflect the characteristics of giant pandas' microhabitat selection. At the same time, the results should try to show the characteristics of microhabitat selection with seasonal differences, rather than listing them all.
Line 237-241: Based on the previous comment, it seems that there is no significant difference in the seasonal selection of giant pandas for the altitude in Niba Mountain area. Instead of listing such meaningless results here.
Line 267-268: The maximum Yodon index values were used to distinguish suitable habitats or unsuitable habitats. the HSI value of suitable habitats in rainy and snowy seasons is too large, so the authors should reconsider the rationality of threshold selection.
Line 330-331: It is recommended to delete the sentence.
Line 428-431: The utilization of the ecological corridor by the giant panda population is determined by a variety of reasons, not only the restoration of vegetation. This conclusion requires careful consideration by the authors.
Lines 451-457: 5. Conclusions need to be rewritten. What is the conclusion of this study? To be clear, the current description seems to have no conclusion here.
Line 471-472: It is recommended deleting the language assistance in the Acknowledgments.
Author Response
Dear Reviewer,Thank you for your comments on our manuscript title "Giant Panda Microhabitat Study in the Daxiangling Niba Mountain Corridor (ID: biology-2123287)". Your comments have been helpful in revising and improving our manuscript and will be an important guide for our future research. We have carefully studied these comments and have worked hard to revise the manuscript in the hope of obtaining your approval. The revisions are marked in red in the manuscript. The following is a response to your comments:
For Main Document:
Comments 1: Line 24: It is recommended that this sentence be deleted from the Abstract.
Response: Dear reviewer, based on your comments, we have deleted this sentence from the Abstract.
Comments 2: Line 37-39: Giant pandas are now distributed in six mountain ranges, and the source of references indexed by them is not representative enough or misquoted. It is suggested to revise it to the latest national census report of giant pandas.Response: Dear reviewer, based on your comments, we have changed the reference to Giant pandas in Sichuan Province: the fourth giant pandas survey report of Sichuan Province. Comments 3: Line 59-60: At least representative articles on giant panda microhabitat selection should be cited.
Response: Dear reviewer, based on your comments, we have added a reference to the paper entitled "Microhabitat selection by giant pandas" published by Bai et al. in 2020.
Comments 4: Line 62-63: It is suggested to point out how the habitat use changes and mark this reference.
Response: Thank you, we have followed your comments and made changes as follows.
Wei et al. showed that between 1999–2000 and 2011–2014, giant pandas in several mountains in Sichuan experienced a shift in habitat use, they have increasingly utilized secondary forest as these forests recovered under protective measures, and have undergone a distributional shift to higher elevations, despite the elevational stability of their bamboo food source.
Comments 5: Line 72-28: The logicality of this paragraph needs to be improved. For example, what is the purpose of evaluating habitat suitability?
Response: Dear reviewer, based on your comments, we have adjusted the logical structure of this paragraph and added the corresponding content. You can see the revised content below.
Determining a suitable habitat distribution pattern within the protected area is essential to formulate a scientific and reasonable giant pandas’ conservation and management plan.
Comments 6: Line 107-112: The time of the sample survey should be clearly stated.
Response: Dear reviewer, based on your comments, we have increased the time of the sample survey. You can see the revised content below.
From October 2020 to April 2022, the data of giant panda traces were recorded using the sample line method and infrared camera monitoring method, and a sample survey was set up in March–April and October–November each year in the areas where giant panda traces were recorded to determine the preferred environmental factors of the giant panda habitat.
Comments 7: Table 2: There is no elevation factor, which seems difficult to understand.
Response: Dear reviewer, based on your comments, we have double-checked this issue, and our results based on the Pearson correlation coefficient of R 4.2.1, variables greater than 0.8 have been deleted, and finally 9 factors have been retained, not including elevation, and the results are shown in the figure S1 below.
Figure S1. Environment variables remaining after removal of high correlation
Comments 8: Line 197-199: The AICc value of the model included in the average can be directly expressed, not just a reference.
Response: Dear reviewer, based on your comments and those of the first reviewer, we have deleted the logistic regression model section because it duplicates the results of the principal component analysis.
Comments 9: 3.1. Results of Giant Panda Habitat Selection Analysis: At least a simple difference test should be added to this part to analyze the seasonal difference of giant panda habitat selection in the study area. The current descriptive statistics cannot reflect the characteristics of giant pandas' microhabitat selection. At the same time, the results should try to show the characteristics of microhabitat selection with seasonal differences, rather than listing them all.
Response: Dear reviewer, based on your comments, we have added the corresponding content. You can see the revised content below.
These habitat selection and ecological niche factor data were input into Excel for the relevant conversions. Following conversion, data were input into SPSS13.0 for normality testing via the one-sample K-S test. Data that conformed to a normal distribution was tested through one-way analysis of variance (ANOVA) and data that did not conform to a normal distribution was tested by the Mann–Whitney U test.
The one-sample K-S test was carried out for 10 ecological factors in the habitat selection plots and control plots. The results showed that six variables—including the altitude, average height of trees, diameter at breast height of trees, tree coverage, average height of shrub, and bamboo height—were normally distributed, while the other 4 variables, including aspect, were not. The plot type was used as the control variable for ANOVA. In normally distributed variables, there were significant differences between the altitude, average height of trees and diameter at breast height of trees (p < 0.05) but no significant differences between the tree coverage, average height of tree, and average height of bamboo (p > 0.05). In the control plots and habitat selection plots, the Mann–Whitney U test disclosed significant differences between two variables, including the shrub coverage and bamboo coverage (p < 0.05). However, there was no significant difference between the aspect and slope (p > 0.05; Table 4).
Table 4 Comparison of variables in habitat selection plots and control plots in the study area by analysis of variance (ANOVA) and Mann–Whitney U tests.
|
Microhabitat variable |
Rainy season |
Snow season |
||
|
ANOVA |
Mann–Whitney U test |
ANOVA |
Mann–Whitney U test |
|
|
F (p) |
U (p) |
F (p) |
U (p) |
|
|
Altitude (m) |
40.04 (0.00***) |
|
36.07 (0.00***) |
|
|
Aspect (°) |
|
171173 (0.17) |
|
165673 (0.08) |
|
Slope (°) |
|
167424 (0.10) |
|
197344 (0.11) |
|
Average height of trees (m) |
35.70 (0.00***) |
|
45.50 (0.00***) |
|
|
Average diameter at breast height of trees (cm) |
9.95 (0.01**) |
|
10.03 (0.01**) |
|
|
Tree coverage (%) |
2.76 (0.97) |
|
2.67(0.93) |
|
|
Average height of shrub (m) |
0.15 (0.70) |
|
0.21 (0.83) |
|
|
Shrub coverage (%) |
|
163905 (0.04*) |
|
164335 (0.04*) |
|
Average height of bamboo (m) |
6.20 (0.13) |
|
6.17 (0.11) |
|
|
Bamboo coverage (%) |
|
146994 (0.00***) |
|
133885 (0.00***) |
Comments 10: Line 237-241: Based on the previous comment, it seems that there is no significant difference in the seasonal selection of giant pandas for the altitude in Niba Mountain area. Instead of listing such meaningless results here.
Response: Dear reviewer, based on your comments, we have deleted the corresponding content and adjusted the sentence structure. You can see the revised content below.
Giant pandas preferred medium altitudes (2000–2400m), southeastern slopes and areas with slopes <15° in both seasons.
Comments 11: Line 267-268: The maximum Yodon index values were used to distinguish suitable habitats or unsuitable habitats. the HSI value of suitable habitats in rainy and snowy seasons is too large, so the authors should reconsider the rationality of threshold selection.
Response: Dear reviewer, based on your comments, we have iteratively verified the accuracy of the model results. We have borrowed this method from Ruan et al. 2022 entitled "Habitat suitability evaluation for giant panda in Liziping National Nature Reserve, Sichuan Province", which is a more scientific method and used the maximum Yodon index to classify the suitable habitat range, and the result is that the maxent model is self-determined and not artificially set.
Comments 12: Line 330-331: It is recommended to delete the sentence.
Response: Dear reviewer, based on your comments, we have deleted this sentence.
Comments 13: Line 428-431: The utilization of the ecological corridor by the giant panda population is determined by a variety of reasons, not only the restoration of vegetation. This conclusion requires careful consideration by the authors.
Response: Dear reviewer, based on your comments, we have deleted the corresponding content. This is because we believe that there are still some factors that have not been considered in the current study results. This is something that needs to be improved and explored in depth in the future.
Comments 14: Lines 451-457:5. Conclusions need to be rewritten. What is the conclusion of this study? To be clear, the current description seems to have no conclusion here.
Response: Dear reviewer, based on your comments, we have rewritten the conclusions and adjusted the sentence structure. You can see the revised content below.
The present study shows that giant pandas prefer to move at medium altitudes; southeast slopes; areas with slopes less than 15°, with overlap of suitable habitats in both seasons. Due to the opening of the G5 Niba Mountain tunnel and the completion of the Niba Mountain giant panda corridor plan, the recovery of vegetation within the Niba Mountain area has led to the emergence of giant panda activity in the area, which may have spread to the central part of the reserve through the corridor. The findings can provide a reference for the future planning and construction of giant panda corridors.
Comments 15: Line 471-472: It is recommended deleting the language assistance in the Acknowledgments.Response: Thank you, we have followed your comments and deleted the language assistance in the Acknowledgments.

Reviewer 3 Report
Overall, there are many sentences in the manuscript that are too long and could be problematic for readers to understand. You can try to follow the criteria given below or you can follow the previously published articles. But each sentence should be very clear and understandable for the readers.
[A. Fairly difficult (Medium long sentence length): 21-25 words. B. Difficult (Long sentence length): 25-30 words. C. Very Difficult (Very long sentence length): 30-40 words. D. Extremely difficult (Extremely long sentence length): 40+ words.]
The whole manuscript needs English improvement. Please ask any native English speaker. Throughout the manuscript, I noticed that mostly you just added a single citation at the end of each sentence/ statement (in the introduction, methodology, and discussion sections) and it’s a bad practice. You need to study more articles and add more citations. I am sure there are a lot of studies on giant pandas in the past for each of your statements. So if you have more citations/references, add them (at least two to three citations but not more than 5).
Summary
1. Line no. 9: Giant pandas should be replaced with “Giant panda throughout the manuscript………”
2. Line no. 11. No need to repeat the species name again and again. Replace “wild giant panda” with “species”.
3. Line no. 12: What is G5? If possible, please explain this G first such as Gap 5 (G5) along with abbreviated G5 in brackets, and later in the whole manuscript, you can use just G5.
4. Line no. 16. Replace “Giant pandas” with “the Giant panda”.
5. Line no. 17-18: What could be the output/ benefits of these finds? Construction of giant panda corridors again sounds not good. Please write down the impact of your study on the conservation/ management of the giant panda.
Abstract
1. Line no. 29-32: This sentence is extremely long. Please break it into at least a couple of sentences.
2. Line no. 31-32: Again, I do not agree with the output of your study findings. Please mention your study’s impact on the conservation/ Management of giant panda in the relevant study area or in the study area in the future.
Keywords
1. Most of the keywords are repetitions of the title except “suitable habitat”. Please go with some different keywords. You can use the most important words that you have used frequently in your study.
Introduction
1. Line no. 38: Replace giant panda with “species”. No need to repeat the word giant panda again and again. Sometimes it's worth saying “species”.
2. Line no. 39: There are a lot of giant panda studies already conducted in these areas. Please add more references here.
3. Line no. 37-41: The first paragraph is very short. It needs a bit more explanation. Prior to the last sentence “Habitat reduction and” in line no. 39 you can add more details. For this, Please read some previously published articles.
4. Line no. 44-46: Please rephrase this sentence.
5. Line no. 46-58: Almost all the sentences are extremely long. Please break each sentence into at least a couple of sentences.
6. Line no. 62: it’s a bad practice to start a sentence with the citation “Wei et al.”. you can cite an article after writing a few words, in the middle or end of the sentence but never in the beginning. Meanwhile, you did not mention the date “Wei et al………..? whats the date of this study?
7. Line no. 67: Same here.
8. Line 67-78: Remove giant pandas with the giant panda.
9. Line no. 72: Remove “To further understand the habitat requirements of giant pandas”. Start as “ In this study, therefore, we analyzed the habitat……………………………
10. Line no. 72-74: This study objective is not very clear. Please rephrase it
11. Line no-78: The study objectives are very important in every study and should be very clear and understandable for the readers what is actually you are going to do in next. But it seems u they were not written properly and comprehensively. Please see the previously published articles and learn how to write objectives in a proper way.
Material and methods:
Study area:
1. Line no. 82. It could be better if you mention the area (in km2) of the Daxiangling Mountains.
2. Line no. 95: Better if add the total area (in km2) for Daxiangling Niba Mountain giant panda ecological corridor also.
3. Line no. 86-88: These sentences indicated the food preferences/ feeding behavior of the giant panda in the area at different elevations. You should move these lines to the introduction part in line no. 70 or anywhere you feel better but make sure it should not make any disturbance in the flow of the whole story. In the study area, you just need to explain the general characteristics of your study area not the food/ feeding of the giant panda.
4. Line no. 86 what do you mean by Daxiangling area? Be consistent, it should be Daxiangling Mountains. Do not use the word “area” again and again same as in line no. 90 “Longcang Valley area”. Write it simply Longcang Valley.
5. Line no. 97: what do you mean by serious destruction?
6. Line no. 100-105: move this paragraph to the section methodology. Where you feel appropriate to place.
7. Line no. 133-148. You did not use any spatial thinning process to minimize the autocorrelation, between the selection of presence points? Why? Please read
8. Dai, Y., Peng, G., Wen, C., Zahoor, B., Ma, X., Hacker, C.E. and Xue, Y., 2021. Climate and land use changes shift the distribution and dispersal of two umbrella species in the Hindu Kush Himalayan region. Science of the Total Environment, 777, p.146207.
9. Zahoor, B., Liu, X., Kumar, L., Dai, Y., Tripathy, B.R. and Songer, M., 2021. Projected shifts in the distribution range of Asiatic black bear (Ursus thibetanus) in the Hindu Kush Himalaya due to climate change. Ecological Informatics, 63, p.101312.
10. Line no. 137: You already mentioned here the buffer you created. So remove it from the study area.
11. Table 2. Just the symbols m or C° are enough. No need to write centigrade and meter in full.
12. Line no. 133-148 and table 2. How you selected these specific bioclimatic and non-bioclimatic variables? How these variables are very important for the distribution of giant panda in the area? Please give a short explanation.
13. Line no 171-173: Rephrase the sentence.
14. Line 185-186: Why you didn’t use R for statistical analysis? Overall, R is the best for statistical analysis also.
15. Line no. 189-196. Very long sentences and poor English, unable to understand.
Results
1. Line no. 204-214. The details of occurrence points have already been described in section 2.3 “Construction of a Suitable Habitat Model for Giant Pandas in the Niba Mountain Corridor”. Why you are repeating almost the same story here?
2. Line no. 237-238. Do you think it’s a very big difference between the elevations between the control and rainy season (So called experimental)? I don’t think just a 6 m difference is a huge and the right way to present your results.
3. Line no. 238-241. How the difference is 60 m between the control sample and the snow sample? The snow season with 2209 m and the control sample with 2074 m?
4. Line no. 242-244: Again long sentence. Further, please rephrase it as the elevation of camera traps does not matter but the elevation of where giant pandas were captured matters. In other words, you can write “the giant pandas were captured (by camera traps) at an elevation of 2165.04 m during rainy season…….and write in the same way for the snowy season.
5. Line no. 245-247: What do you mean by “The average slope in the rainy season (158.27°) and the snow season (141.39°) were both at the southeast slope, and the average slope in the rainy season (12.81°) and the snow season (8.89°) were both less than 15°”? I Could not understand.
6. Line no. 249: Taller, larger? It doesn’t make sense. You should categorize it as tall and taller or large and larger…………… Taller and larger are synonyms. How you can use it?
7. Line no. 249-251: How you categorized “moderately dense tree layers, shorter (<4 m) and sparse (><30%) shrub layers, and taller (>2 m) shrub layers and denser (>35%) 250 bamboos”? and where you described these in the methodology section?
8. Line no. 297: “As shown in Table 3” is not a good way to start a sentence. Please see previously published articles.
9. Overall, I could not understand most of the text in the results section. Improvement in English is much needed.
Discussion
1. Line no. 330-331: Just a couple of lines in the very first paragraph? You need to explain a bit more according to your study findings. It is not a good way to write a couple of lines from any previously published article and then started a new heading.
2. Line no. 333: Again started with a citation and no date…Very poor practice of writing. You need to read some latest publications of good journals on the giant panda and even other species.
3. Line no. 333-336. This text should move into the introduction part just before the objectives. You can write it as a study gap. what they’ve already done and what they couldn’t. Then you can write that, now this is something new that you’re going to do… then you can write objectives….
4. Line 340. “From Figure 1” is again not a good way to start a sentence. Further “it can be seen” is not a proper way …You can write “The traces of giant panda activities were recorded during different seasons in the Niba Mountain corridor (Fig. 1).
5. Line no. 342-343: “so the planning of the giant panda corridor in Niba Mountain is reasonable.” What does it mean?
6. Line no. 352-353: “Giant panda activities have been recorded in the Niba Mountain area since 2016” repetition of line no. 349-350.
7. Line no. 354: Please double check with” bamboo shooting”. I am not sure if it is an exact term.
8. Line no. 358: Did same. Started with Citations………..
9. Line no. 358-362: These lines also need to move in the introduction section before objectives as a research gap. Further, the sentence is extremely long
10. Discussion is something else. You need to discuss your main findings. You can include the previous studies just to prove/disprove or strengthen your findings. Discussion and introduction are the two most important parts of an article. Please learn how to write a good discussion and a good intro. Don’t just repeat the introduction and results in the discussion.
11. Line no. 362-363: “The results of the analysis of giant panda habitat selection in the present study showed”. Write in a simple way……..Our results indicated that/ We explored that/ our study revealed that/ Our study showed that/ Present study showed that/ Present study indicated that etc……………….
12. Line no. 365-366: “which was similar to the results reported by Fu et al. and Bai et al [1,34].” You can write “as reported by Fu et al. and Bai et al [1,34].”
13. Line no. 370. Please add some references here even a single reference is okay.
14. Line no. 371-373: Cross-check with your results after making corrections.
15. Line no 386-387: What do you mean by “It is suggested”? why you used the word suggested? And who suggested it?
16. Line no. 391: what do you mean by ”narrower”? do you mean reduced?
17. Line no. 395-397: rephrase this sentence.
18. Line no. 399-401 “This statement is a repetition of line no. 390-392.
19. Line no. 405-406: “most suitable, suitable, more suitable, less suitable, and unsuitable? This should be as “most suitable, more suitable, suitable, less suitable, and unsuitable”. Or even you can categorize it as “most suitable, suitable, less suitable, and unsuitable” and this is a much better way I think.
20. Line no. 413: “and the placement of infrared cameras began” rephrase it.
21. Line no. 421-425: Extremely long sentence. Also, need references.
22. Line no. 425-428. Extremely long sentence. Also please rephrase it.
23. Line no. 432-438: I like that you discussed future directions. What researchers or even you can do further in the future. This is an important part of the discussion. But again I would say that the way you discussed it here I don’t like. Please rephrase. Furthermore, this whole paragraph is just one sentence, and its a very very very extremely long sentence.
24. So no suggestions or recommendations at the end of the discussion? Why?
Author Response
Thank you for your comments on our manuscript title " Giant Panda Microhabitat Study in the Daxiangling Niba Mountain Corridor (ID: biology-2123287)". Your comments have been helpful in revising and improving our manuscript and will be an important guide for our future research. We have carefully studied these comments and have worked hard to revise the manuscript in the hope of obtaining your approval. The revisions are marked in red in the manuscript. The following is a response to your comments:
For Main Document:
Comments 1: Line no. 9: Giant pandas should be replaced with “Giant panda throughout the manuscript………”
Response: Dear reviewer, based on your comments, we have replaced “Giant pandas” with “Giant panda” throughout the manuscript.
Comments 2: Line no. 11. No need to repeat the species name again and again. Replace “wild giant panda” with “species”. Response: Dear reviewer, based on your comments, we have replaced “wild giant panda” with “species”. Comments 3: Line no. 12: What is G5? If possible, please explain this G first such as Gap 5 (G5) along with abbreviated G5 in brackets, and later in the whole manuscript, you can use just G5.
Response: Dear reviewer, G5 means the National Road 5, which we have explained in the manuscript.
Comments 4: Line no. 16. Replace “Giant pandas” with “the Giant panda”.
Response: Dear reviewer, based on your comments, we have replaced “Giant pandas” with “the Giant panda”
Comments 5: Line no. 17-18: What could be the output/ benefits of these finds? Construction of giant panda corridors again sounds not good. Please write down the impact of your study on the conservation/ management of the giant panda.
Response: Dear reviewer, based on your comments, we have rewritten the corresponding content. You can see the revised content below.
These findings can provide a reference for scientists to formulate practical habitat conservation and management measures for the species in the study area.
Abstract
Comments 1: Line no. 29-32: This sentence is extremely long. Please break it into at least a couple of sentences.
Response: Dear reviewer, based on your comments, we have rewritten the corresponding content.
Comments 2: Line no. 31-32: Again, I do not agree with the output of your study findings. Please mention your study’s impact on the conservation/ Management of giant panda in the relevant study area or in the study area in the future.
Response: Dear reviewer, based on your comments, we have rewritten the corresponding content. You can see the revised content below.
The findings of the study can provide scientific references for the development of practical habitat conservation and management measures for the giant panda in the study area.
KeywordsComments 1: Most of the keywords are repetitions of the title except “suitable habitat”. Please go with some different keywords. You can use the most important words that you have used frequently in your study.
Response: Dear reviewer, based on your comments, we have changed some keywords. You can see the revised content below.
Keywords: Ailuropoda melanoleuca; Niba Mountain corridor; suitable habitat; habitat selection; Principal component analysis
IntroductionComments 1: Line no. 38: Replace giant panda with “species”. No need to repeat the word giant panda again and again. Sometimes it's worth saying “species”.
Response: Dear reviewer, based on your comments, we have rewritten the corresponding content.
Comments 2: Line no. 39: There are a lot of giant panda studies already conducted in these areas. Please add more references here.
Response: Dear reviewer, based on your comments, we have added references to giant panda studies in these areas.
Comments 3: Line no. 37-41: The first paragraph is very short. It needs a bit more explanation. Prior to the last sentence “Habitat reduction and” in line no. 39 you can add more details. For this, please read some previously published articles.
Response: Dear reviewer, based on your comments, we have added details on the decline of the giant panda habitat and the corresponding references. You can see the revised content below.
In the past decades, the giant panda population has decreased, with suitable habitat areas becoming increasingly shrunken and fragmented due to an imbalance between economic development and ecological conservation.
Comments 4: Line no. 44-46: Please rephrase this sentence.
Response: Dear reviewer, based on your comments, we have rephrased the sentence.
Comments 5: Line no. 46-58: Almost all the sentences are extremely long. Please break each sentence into at least a couple of sentences.
Response: Dear reviewer, based on your comments, we have rewritten the corresponding content.
Comments 6: Line no. 62: it’s a bad practice to start a sentence with the citation “Wei et al.”. you can cite an article after writing a few words, in the middle or end of the sentence but never in the beginning. Meanwhile, you did not mention the date “Wei et al………? whats the date of this study?
Response: Dear reviewer, based on your comments, we have rewritten the corresponding content. You can see the revised content below.
Previous studies showed that between 1999–2000 and 2011–2014, giant panda in several mountains in Sichuan experienced a shift in habitat use, they have increasingly utilized secondary forest as these forests recovered under protective measures, and have undergone a distributional shift to higher elevations, despite the elevational stability of their bamboo food source.
Comments 7: Line no. 67: Same here.
Response: Dear reviewer, based on your comments, we have rewritten the corresponding content. You can see the revised content below.
Previous studies reported changes in the habitat selection of giant panda in the Daxiangling area between 2001 and 2020 and found that highly edible bamboo and good shelter sites had an important influence on the habitat selection of giant panda.
Comments 8: Line 67-78: Remove giant pandas with the giant panda.
Response: Dear reviewer, based on your comments, we have removed giant pandas with the giant panda.
Comments 9: Line no. 72: Remove “To further understand the habitat requirements of giant pandas”. Start as “In this study, therefore, we analyzed the habitat……………………………
Response: Dear reviewer, based on your comments, we have rewritten the corresponding content. You can see the revised content below.
In this study, therefore, we analyzed the habitat selection characteristics and differences of the giant panda based on the sample data of giant panda occurrence sites collected during 2020–2022.
Comments 10: Line no. 72-74: This study objective is not very clear. Please rephrase it
Response: Dear reviewer, based on your comments and those of the second reviewer, we have clarified the objective of the study. You can see the revised content below.
Determining a suitable habitat distribution pattern and selection characteristics within the protected area is essential to formulate a scientific and reasonable giant panda conservation and management plan.
Comments 11: Line no-78: The study objectives are very important in every study and should be very clear and understandable for the readers what is actually you are going to do in next. But it seems u they were not written properly and comprehensively. Please see the previously published articles and learn how to write objectives in a proper way.
Response: Dear reviewer, based on your comments, we have clarified the objective of the study and rewritten them correspondingly. You can see the revised content below.
Determining a suitable habitat distribution pattern and selection characteristics within the protected area is essential to formulate a scientific and reasonable giant panda conservation and management plan. In this study, therefore, we analyzed the habitat selection characteristics and differences of the giant panda based on the sample data of giant panda occurrence sites collected during 2020–2022.
Material and methodsStudy area:Comments 1: Line no. 82. It could be better if you mention the area (in km2) of the Daxiangling Mountains.
Response: Dear reviewer, based on your comments, we have added the area of the Daxiangling Mountains. You can see the revised content below.
The Daxiangling Mountains are located in the transition zone between the Sichuan Basin and the Qinghai–Tibet Plateau in the eastern part of the Hengduan Mountains, covering an area of about 6440 km².
Comments 2: Line no. 95: Better if add the total area (in km2) for Daxiangling Niba Mountain giant panda ecological corridor also.
Response: Dear reviewer, based on your comments, we have added the area of the Daxiangling Niba Mountain giant panda ecological corridor. You can see the revised content below.
The Daxiangling Niba Mountain giant panda ecological corridor (102°29′–102°52′E, 29°28′–29°43′N) is located in the southwestern part of the reserve in Yingjing County and Hanyuan County, Sichuan Province, covering an area of about 38.5 km².
Comments 3: Line no. 86-88: These sentences indicated the food preferences/ feeding behavior of the giant panda in the area at different elevations. You should move these lines to the introduction part in line no. 70 or anywhere you feel better but make sure it should not make any disturbance in the flow of the whole story. In the study area, you just need to explain the general characteristics of your study area not the food/ feeding of the giant panda.
Response: Dear reviewer, based on your comments, we have moved these lines to the introduction part. You can see the revised content below.
Comments 4: Line no. 86 what do you mean by Daxiangling area? Be consistent, it should be Daxiangling Mountains. Do not use the word “area” again and again same as in line no. 90 “Longcang Valley area”. Write it simply Longcang Valley.
Response: Dear reviewer, based on your comments, we have rewritten the corresponding content.
Comments 5: Line no. 97: what do you mean by serious destruction?
Response: Dear reviewer, based on your comments, we have added the specific meaning of serious destruction. You can see the revised content below.
The vegetation on both sides of the national road is seriously destructed, and many trees and bamboo for pandas have been cut down due to road construction.
Comments 6: Line no. 100-105: move this paragraph to the section methodology. Where you feel appropriate to place.
Response: Dear reviewer, based on your comments, we have moved the corresponding content.
Comments 7: Line no. 133-148. You did not use any spatial thinning process to minimize the autocorrelation, between the selection of presence points? Why? Please read
Response: Dear reviewer, we have carefully read the references you have recommended. Because we have operated through the function of generating buffers in ArcGIS 10.2. A buffer zone is generated from the giant panda occurrence site centered on a general radius of 1125 m of giant panda activity. When the locus buffers overlap each other, one of them is kept randomly and the rest are eliminated. This method was referred from other articles doing habitat selection. Because there were fewer occurrence sites in the rainy and snow seasons and the overlap was low, the redundancy analysis was done and found to be not very different, so the redundancy analysis steps were not listed separately.
Comments 8: Dai, Y., Peng, G., Wen, C., Zahoor, B., Ma, X., Hacker, C.E. and Xue, Y., 2021. Climate and land use changes shift the distribution and dispersal of two umbrella species in the Hindu Kush Himalayan region.Science of the Total Environment,777, p.146207.
Response: Dear reviewer, we have carefully read the references you have recommended.
Comments 9: Zahoor, B., Liu, X., Kumar, L., Dai, Y., Tripathy, B.R. and Songer, M., 2021. Projected shifts in the distribution range of Asiatic black bear (Ursus thibetanus) in the Hindu Kush Himalaya due to climate change.Ecological Informatics,63, p.101312.
Response: Dear reviewer, we have carefully read the references you have recommended.
Comments 10: Line no. 137: You already mentioned here the buffer you created. So remove it from the study area.
Response: Dear reviewer, based on your comments, we have removed the buffer contents from the study area.
Comments 11: Table 2. Just the symbols m or C° are enough. No need to write centigrade and meter in full.
Response: Dear reviewer, based on your comments, we have rewritten the corresponding content.
Comments 12: Line no. 133-148 and table 2. How you selected these specific bioclimatic and non-bioclimatic variables? How these variables are very important for the distribution of giant panda in the area? Please give a short explanation.
Response: Dear reviewer, based on your comments, we have added explanations for selecting various environment variables. You can see the revised content below.
The giant pandas have rigorous requirements for habitat, usually choosing primary forest with low interference. Climate and land-use type are also important factors that influence the spatial distribution of giant pandas. Therefore, climate, topography, vegetation, and human disturbance are important factors affecting the spatial distribution of giant panda.
Comments 13: Line no 171-173: Rephrase the sentence.
Response: Dear reviewer, based on your comments, we have rewritten the corresponding content. You can see the revised content below.
The means of 10 calculation results were analyzed to gain the habitat suitability index (HSI). The suitable habitat range in the study area was divided, using Youden’s index as the threshold.
Comments 14: Line 185-186: Why you didn’t use R for statistical analysis? Overall, R is the best for statistical analysis also.
Response: Dear reviewer, because we also used SPSS to do principal component analysis before, which is more familiar to us and the method is more scientific, we will also try to do it with R in the future.
Comments 15: Line no. 189-196. Very long sentences and poor English, unable to understand.
Response: Dear reviewer, I am very sorry to show you such a bad English expression, based on your comments and those of the first reviewer, we have removed the logistic Stiff model part of the R language as it appeared to duplicate the results of the principal component analysis.
ResultsComments 1: Line no. 204-214. The details of occurrence points have already been described in section 2.3 “Construction of a Suitable Habitat Model for Giant Pandas in the Niba Mountain Corridor”. Why you are repeating almost the same story here?
Response: Dear reviewer, based on your comments, we have rewritten the corresponding content. You can see the revised content below.
The giant panda occurrence sites included 18 occurrence sites recorded using the sample line method and 26 occurrence sites photographed by infrared cameras in the rainy season. All occurrence sites were photographed by infrared cameras during the snow season. The control samples were 155 and 131 in the rainy and snow seasons, respectively (Table 3).
Comments 2: Line no. 237-238. Do you think it’s a very big difference between the elevations between the control and rainy season (So called experimental)? I don’t think just a 6 m difference is a huge and the right way to present your results.
Response: Dear reviewer, based on your comments, we have rewritten the corresponding content. You can see the revised content below.
Giant panda preferred medium altitudes (2000–2400m), southeastern slopes and areas with slopes <15° in both seasons.
Comments 3: Line no. 238-241. How the difference is 60 m between the control sample and the snow sample? The snow season with 2209 m and the control sample with 2074 m?
Response: Dear reviewer, based on your comments, we have rewritten the corresponding content. You can see the revised content below.
Giant panda preferred medium altitudes (2000–2400m), southeastern slopes and areas with slopes <15° in both seasons.
Comments 4: Line no. 242-244: Again long sentence. Further, please rephrase it as the elevation of camera traps does not matter but the elevation of where giant pandas were captured matters. In other words, you can write “the giant pandas were captured (by camera traps) at an elevation of 2165.04 m during rainy season…….and write in the same way for the snowy season.
Response: Dear reviewer, based on your comments, we have rewritten the corresponding content. You can see the revised content below.
The giant pandas were captured (by camera traps) at an average elevation of 2165.04 m during the rainy season, and 2209.17 m during the snow season. Both are at medium elevation, and the average elevation in the snow season was about 44 m higher than in the rainy season.
Comments 5: Line no. 245-247: What do you mean by “The average slope in the rainy season (158.27°) and the snow season (141.39°) were both at the southeast slope, and the average slope in the rainy season (12.81°) and the snow season (8.89°) were both less than 15°”? I Could not understand.
Response: Dear reviewer, based on your comments, we have rewritten the corresponding content. You can see the revised content below.
The giant pandas were captured at an average aspect of 158.27° during the rainy season, and 141.39° during the snow season. Both were at the southeast slope. And the giant pandas were captured at an average slope of 12.81° during the rainy season, and 8.89° during the snow season. Both were less than 15°.
Comments 6: Line no. 249: Taller, larger? It doesn’t make sense. You should categorize it as tall and taller or large and larger…………… Taller and larger are synonyms. How you can use it?
Response: Dear reviewer, based on your comments, we have rewritten the corresponding content. You can see the revised content below.
The preferred community structure of the giant panda habitat was characterized by a preference for tall (8–15 m), large (17–25 cm) diameter at breast height and moderately dense (25–55%) tree layers, short (<4 m) and sparse (<30%) shrub layers, and tall (>2 m) and dense (>35%) bamboos in both seasons.
Comments 7: Line no. 249-251: How you categorized “moderately dense tree layers, shorter (<4 m) and sparse (><30%) shrub layers, and taller (>2 m) shrub layers and denser (>35%) 250 bamboos”? and where you described these in the methodology section?
Response: Dear reviewer, based on your comments, we have added the classification criteria for the different environment variables as shown in Table 1. You can see the revised content below.
The classification criteria for different environmental variables are shown in Table 1.
Table 1. Classification criteria for microhabitat variables
|
Microhabitat variable |
Classification criteria |
|
Altitude |
<1 400 m (evergreen broadleaf forests); 1 400 -1 800 m (deciduous broadleaf forests); 1 800 -2 600 m (mixed coniferous forests); 2 600 -3 100 m (coniferous forests); >3 100 m (alpine scrub meadows) |
|
Aspect |
<22.5°, >337.5°(N); 22.5°-67.5°(NE); 67.5°-112.5°(E); 112.5°-157.5°(SE); 157.5°-202.5°(S); 202.5°-247.5°(SW); 247.5°-292.5°(W); 292.5°-337.5°(NW) |
|
Slope |
<15°; 15°-30°; >30° |
|
Average height of trees |
<8 m; 8 -12 m; >12 m |
|
Average diameter at breast height of trees |
<17 cm; 17 -25 cm; >25 cm |
|
Tree coverage |
<25 %; 25 %-55 %; >55 % |
|
Shrub height |
<3 m; 3 -4 m; >4 m |
|
Shrub coverage |
<30 %; 30 %-50 %; >50 % |
|
Average height of bamboo |
<1 m; 1 -2 m; >2 m |
|
Bamboo coverage |
<20 %; 20 %-35 %; >35 % |
Comments 8: Line no. 297: “As shown in Table 3” is not a good way to start a sentence. Please see previously published articles.
Response: Dear reviewer, based on your comments and the format of previously published articles, we have rewritten the corresponding content.
Comments 9: Overall, I could not understand most of the text in the results section. Improvement in English is much needed.
Response: Dear reviewer, I am very sorry to show you such a bad English expression, we have sent the article to a language retouching company for re-touching. Thank you so much!
DiscussionComments 1: Line no. 330-331: Just a couple of lines in the very first paragraph? You need to explain a bit more according to your study findings. It is not a good way to write a couple of lines from any previously published article and then started a new heading.
Response: Dear reviewer, based on your comments and those of the second reviewer, we have removed the corresponding content because it did not feel relevant to the context and seemed redundant.
Comments 2: Line no. 333: Again started with a citation and no date…Very poor practice of writing. You need to read some latest publications of good journals on the giant panda and even other species.
Response: Dear reviewer, based on your comments, we have added the following to the end of the abstract and would appreciate your approval.
Comments 3: Line no. 333-336. This text should move into the introduction part just before the objectives. You can write it as a study gap. what they’ve already done and what they couldn’t. Then you can write that, now this is something new that you’re going to do… then you can write objectives….
Response: Dear reviewer, based on your comments and those of the first reviewer, we have rewritten the corresponding content. You can see the revised content below.
Previous studies have investigated the population and habitat of giant panda in Daxiangling and confirmed the distribution of giant panda in Yingjing County, Daxiangling Reserve, and the Niba Mountain area, respectively, but did not specify the main activity area of the giant panda
Comments 4: Line 340. “From Figure 1” is again not a good way to start a sentence. Further “it can be seen” is not a proper way …You can write “The traces of giant panda activities were recorded during different seasons in the Niba Mountain corridor (Fig. 1).
Response: Dear reviewer, based on your comments, we have rewritten the corresponding content. You can see the revised content below.
The traces of giant panda activities were recorded during different seasons in the Niba Mountain corridor (Figure 1). Furthermore, the species spread in the middle of the corridor. and the planning of the giant panda corridor in Niba Mountain is reasonable.
Comments 5: Line no. 342-343: “so the planning of the giant panda corridor in Niba Mountain is reasonable.” What does it mean?
Response: Dear reviewer, based on your comments, we have added the corresponding content to explain why the construction of the Niba Mountain corridor is reasonable. You can see the revised content below.
Therefore, giant panda may have been able to use the corridor to migrate, and the planning of the giant panda corridor in Niba Mountain is reasonable.
Comments 6: Line no. 352-353: “Giant panda activities have been recorded in the Niba Mountain area since 2016” repetition of line no. 349-350.
Response: Dear reviewer, based on your comments, we have rewritten the corresponding content. You can see the revised content below.
The Niba Mountain has become an area with frequent giant panda activity.
Comments 7: Line no. 354: Please double check with” bamboo shooting”. I am not sure if it is an exact term.
Response: Dear reviewer, based on your comments, we have reviewed previous references and replaced “bamboo shooting” with “bamboo shoot collection”.
Comments 8: Line no. 358: Did same. Started with Citations………..
Response: Dear reviewer, based on your comments, we have rewritten the corresponding content. You can see the revised content below.
Previous studies have also preliminarily explored the habitat selection of giant panda in the Daxiangling Mountains.
Comments 9: Line no. 358-362: These lines also need to move in the introduction section before objectives as a research gap. Further, the sentence is extremely long.
Response: Dear reviewer, based on your comments, we have split this sentence and moved the corresponding content to the introduction part. You can see the revised content below.
Previous studies have also preliminarily explored the habitat selection of giant panda in the Daxiangling Mountains. However, these studies only described the overall habitat selection of giant panda throughout the year. The habitat selection and utilization of giant panda under the influence of different environmental factor conditions in different seasons were not further elaborated.
Comments 10: Discussion is something else. You need to discuss your main findings. You can include the previous studies just to prove/disprove or strengthen your findings. Discussion and introduction are the two most important parts of an article. Please learn how to write a good discussion and a good intro. Don’t just repeat the introduction and results in the discussion.
Response: Dear reviewer, based on your comments, we have rechecked the content of the introduction and discussion section, removed some repetitive and irrelevant words, and highlighted the results of our research.
Comments 11: Line no. 362-363: “The results of the analysis of giant panda habitat selection in the present study showed”. Write in a simple way……..Our results indicated that/ We explored that/ our study revealed that/ Our study showed that/ Present study showed that/ Present study indicated that etc………………
Response: Dear reviewer, based on your comments, we have rewritten the corresponding content. You can see the revised content below.
Our study showed that there were distinctive features of giant panda habitat selection in the study area.
Comments 12: Line no. 365-366: “which was similar to the results reported by Fu et al. and Bai et al [1,34].” You can write “as reported by Fu et al. and Bai et al [1,34].”
Response: Dear reviewer, based on your comments, we have rewritten the corresponding content. You can see the revised content below.
Giant panda chose to move in southeastern gently sloping areas at medium altitude with tall trees in both seasons as reported by Fu et al. and Bai et al.
Comments 13: Line no. 370. Please add some references here even a single reference is okay.
Response: Dear reviewer, based on your comments, we have added some appropriate references.
Comments 14: Line no. 371-373: Cross-check with your results after making corrections.
Response: Dear reviewer, based on your comments, we have Cross-checked our results after making corrections and rewritten the corresponding content. You can see the revised content below.
Giant panda preferred to move at moderate altitudes in both seasons, but the average distribution elevation in the snow season was about 62 m higher than that in the rainy season.
Comments 15: Line no 386-387: What do you mean by “It is suggested”? why you used the word suggested? And who suggested it?
Response: Dear reviewer, we used "suggest" to mean indicate, not suggest. Sorry for the misunderstanding of the word, we have changed it to the synonym "indicate". You can see the revised content below.
Our results indicated that altitude affects the bamboo shoot collection, the staple food of giant panda, and thus influences the habitat selection of giant panda in different seasons.
Comments 16: Line no. 391: what do you mean by ”narrower”? do you mean reduced?
Response: Dear reviewer, based on your comments, there may be a mistake in our expression, We have rewritten the corresponding content. You can see the revised content below.
The distribution of suitable giant panda habitat was more fragmented in the snow season than in the rainy season.
Comments 17: Line no. 395-397: rephrase this sentence.
Response: Dear reviewer, based on your comments, we have rephrased this sentence. You can see the revised content below.
The suitable habitat area for giant panda in the snow season overlapped with that during the rainy season. The overlapping area is about 26.85 km2, which accounts for 24.34 % of the total suitable habitat area.
Comments 18: Line no. 399-401 “This statement is a repetition of line no. 390-392.
Response: Dear reviewer, based on your comments, we have removed the corresponding content.
Comments 19: Line no. 405-406: “most suitable, suitable, more suitable, less suitable, and unsuitable? This should be as “most suitable, more suitable, suitable, less suitable, and unsuitable”. Or even you can categorize it as “most suitable, suitable, less suitable, and unsuitable” and this is a much better way I think.
Response: Dear reviewer, based on your comments, we have rewritten the corresponding content. You can see the revised content below.
The next step will be to introduce a finer level (four levels: most suitable, suitable, less suitable, and unsuitable) of assessment criteria for giant panda habitat assessment in the study area to explore the potential habitat and dispersal pathways of giant panda in the study area.
Comments 20: Line no. 413: “and the placement of infrared cameras began” rephrase it.
Response: Dear reviewer, based on your comments, we have removed the corresponding content. You can see the revised content below.
A giant panda activity trail was discovered in the central part of the Niba Mountain corridor in 2019. Two infrared cameras capturing giant panda activity in February 2022.
Comments 21: Line no. 421-425: Extremely long sentence. Also, need references.
Response: Dear reviewer, based on your comments, we have rewritten the corresponding content. You can see the revised content below.
Therefore, it is suggested that another ecological corridor for the species can be planned from the central part of the reserve to the Longcang Valley. Alternatively, the eastern part of the Niba Mountain corridor can be extended to the Longcang Valley to strengthen the exchange of giant panda populations.
Comments 22: Line no. 425-428. Extremely long sentence. Also please rephrase it.
Response: Dear reviewer, based on your comments, we have rewritten the corresponding content. You can see the revised content below.
The newly recorded point of giant panda activity was found in Niba Mountain in 2016 and the newly recorded point of giant panda activity was found in the middle of the Niba Mountain corridor in 2019. This indicates that it takes about 3 years for the giant panda to move from the west to the east of the reserve and from the north to the south.
Comments 23: Line no. 432-438: I like that you discussed future directions. What researchers or even you can do further in the future. This is an important part of the discussion. But again I would say that the way you discussed it here I don’t like. Please rephrase. Furthermore, this whole paragraph is just one sentence, and its a very very very extremely long sentence.
Response: Dear reviewer, thank you so much! Based on your comments, we have rewritten the corresponding content. You can see the revised content below.
We are trying to better verify the actual role of the Niba Mountain giant panda corridor in the dispersal of the giant panda. In the next step, biological samples such as fresh feces will be collected from giant panda in different areas. We will explore the genetic diversity of the giant panda populations by combining microsatellite and mitochondrial control regions using feces as the main experimental material. Further, we will explore the origin of newly recorded giant panda occurrence sites in the central part of the country. To determine the dispersal path of the giant panda population in the Daxiangling Mountains.
Comments 24: So no suggestions or recommendations at the end of the discussion? Why?
Response: Dear reviewer, it is not that we did not want to add a suggestion at that time, because we felt that our current work is only a preliminary study. Further research is needed to get more scientific conclusions and give more practical and feasible suggestions or recommendations. So we wrote a suggestion at that time but finally deleted it. Based on your comments, we have added the suggestions. You can see the revised content below.
We recommend monitoring giant pandas near roads and their use of corridors (e.g., road tunnels) to evaluate their impact on giant pandas. Further research should assess the effects of bamboo cover on the foraging and movement of giant pandas. Meanwhile, it is suggested that field patrols be strengthened to decrease human activities in the giant panda’s habitat, including grazing, herb collection, bamboo shoot collection, and hunting.

Round 2
Reviewer 2 Report
I have no comments or suggestions on the revised manuscript.
Author Response
Thanks a lot of your hard work on this manuscript, and hope you could help us to improve the study work in the future. We appreciate your comments for this manuscript in the round one.
Reviewer 3 Report
1. The English of the manuscript is worse in its current form. This Manuscript could be acceptable after English Improvement. Further, the author should submit the final version with the changes below:
2. Please move line no. 80-81 and merge it with line no. 75-76.
3. still this manuscript does not have very clear objectives.
4. Line no. 407-412: So only three recommendations here? Nothing more? If possible you can add 2-3 more recommendations here.
Author Response
Dear Reviewer,Thank you for your comments on our manuscript entitled "Giant Panda Microhabitat Study in the Daxiangling Niba Mountain Corridor (ID: biology-2123287)". Your comments have been helpful in revising and improving our manuscript and will be an important guide for our future research. We have carefully studied these comments and have worked hard to revise the manuscript in the hope of obtaining your approval. The revisions are marked in red in the manuscript. Point-by-point responses to your comments are provided below.
For the Main Document:
Comment 1: The English of the manuscript is worse in its current form. This Manuscript could be acceptable after English Improvement. Further, the author should submit the final version with the changes below:
Response: Dear reviewer, I apologize for the errors. We have sent the manuscript to a language editing company for polishing. Thank you so much for your careful review!
Comment 2: Please move line no. 80-81 and merge it with line no. 75-76.
Response: Dear reviewer, we have rewritten the corresponding content based on your comments. You can see the revised content below.
Previous studies have also investigated the giant panda's habitat selection in the Daxiangling Mountains and reported changes in the habitat selection of this species between 2001 and 2020. Comment 3: still this manuscript does not have very clear objectives.
Response: Dear reviewer, we have added clear objectives to the introduction based on your comments. You can see the revised content below.
Therefore, based on the sample data of giant panda occurrence sites collected during 2020–2022, this study aimed to apply MaxEnt and other methods to evaluate the habitat selection characteristics, including the spatial distribution pattern of suitable habitats for the giant panda in the Daxiangling Niba Mountain corridor. The research conclusions can provide a scientific reference for giant panda habitat conservation and management in the area.
Comment 4: Line no. 407-412: So only three recommendations here? Nothing more? If possible you can add 2-3 more recommendations here.
Response: Dear reviewer, we have added some recommendations based on your comments. You can see the revised content below.
We recommend monitoring the movement of the giant panda near roads and their use of corridors (e.g., road tunnels) to evaluate the impact of corridors on the giant panda. Further research should assess the effects of bamboo cover on the foraging and movement of the giant panda. In addition, it is suggested that field patrols should be strengthened to decrease human activities in the giant panda habitat, including grazing, herb collection, bamboo shoot collection, and hunting. The future production and living modes of residents can be driven by ecological tourism development and the exploration of agricultural and sideline products with regional characteristics, which may change the negative attitudes of residents toward ecological conservation. This method can promote industrial transformation to realize the balanced development of economic construction and ecological conservation. Additionally, the Daxiangling Mountains should focus on the continuous conservation of major habitats for the giant panda, decrease habitat fragmentation caused by anthropogenic interference, and strengthen habitat restoration for the giant panda
